# Design, Optimization, and Evaluation of Additively Manufactured Vintiles Cellular Structure for Acetabular Cup Implant

**Kalayu Mekonen Abate [1,2], Aamer Nazir [1,2], Jia-En Chen [3] and Jeng-Ywan Jeng [1,2,*]**

1   High Speed 3D Printing Research Center, National Taiwan University of Science and Technology, #43, Sec.4, Keelung Road, Taipei 106, Taiwan; d10503819@mail.ntust.edu.tw (K.M.A.); d10503822@mail.ntust.edu.tw (A.N.)
2   Department of Mechanical Engineering, National Taiwan University of Science and Technology, No. 43, Section 4, Keelung Road, Taipei 106, Taiwan
3   Department of Biomedical Engineering, National Defense Medical Center, Taipei 11490, Taiwan; w112233442@gmail.com
*   Correspondence: jeng@mail.ntust.edu.tw

**Abstract:** Cellular materials with very highly regulated micro-architectures are promising applicant materials for orthopedic medical uses while requiring implants or substituting for bone due to their ability to promote increased cell proliferation and osseointegration. This study focuses on the design of an acetabular cup (AC) cellular implant which was built using a vintiles cellular structure with an internal porosity of 56–87.9% and internal pore dimensions in the range of 600–1200 μm. The AC implant was then optimized for improving mechanical performance to reduce stress shielding by adjusting the porosity to produce stiffness (elastic modulus) to match with the bone, and allowing for bone cell ingrowth. The optimized and non-optimized AC cellular implant was fabricated using the SLM additive manufacturing process. Simulation (finite element analysis, FEA) was carried out and all cellular implants are finally tested under static loading conditions. The result showed that on the finite element model of an optimized implant, cellular has shown 69% higher stiffness than non-optimized. It has been confirmed by experimental work shown that the optimized cellular implant has a 71% higher ultimate compressive strength than the non-optimized counterpart. Finally, we developed an AC implant with mechanical performance adequately close to that of human bone.

**Keywords:** cellular structure; acetabular cup; cellular implant; additive manufacturing; finite element analysis; mechanical property; design and optimization

## 1. Introduction

Cellular structures are abundant in nature and have been broadly used in engineering and biomedical applications such as sound and energy absorbers, lightweight structural components, heat transfer, implants, and scaffolds for years [1]. Commonly, cellular structures can be categorized into two groups: Periodic cellular structures (lattice structures) and stochastic foams (honeycombs). Cellular structures categorized by open unit cells non-stochastic have better mechanical performances in contrast to stochastic foams which show localized deformations caused by internal imperfections [2,3]. Periodic cellular structures have certain combinations of geometrical features physical and mechanical performance that made them appropriate for many applications such as in medical and other major industries [4]. Recently, cellular structures have been broadly studied as promising applicants for implant (biomedical) use such as bone grafting or osseointegration [5,6] due to their comparable properties and inner topological complexity to the host bone. Since bone growth depends on

morphological characteristics such as permeability, pore size, volume-surface ratio, and pore architecture by regulating the morphological parameters, it can facilitate bone ingrowth, cell migration, and achieve tailor biomechanical properties [7,8]. Yan et al. investigated cellular implants which have a similar mechanical property with human bone by regulating the morphology of porosity and elastic modulus with low stress-shielding effect [9]. Melchels et al. investigated the effect of scaffold pore shape on static culturing and cell seeding and revealed that the gyroid cellular structure has a high permeability than salt leaching architecture by improving morphology parameters [10].

Several researchers have investigated various cellular structures specifically for biomedical applications. Among the various cellular structures, a 2D lattice-based geometry is designed for biomaterial made of two-dimensional unit cells, for example, honeycombs [11,12]. On the other side, 3D lattice-based geometries, such as diamond lattice topology is a suitable candidate cellular structure for orthopedic application [13,14]. Using unit-cell tetrakaidecahedron developed an open-celled cellular (foam) for the biomedical implant and determined the mechanical property at the macrostructure level such as young's modulus and Poisson's ratio [15–17]. The tetrahedron is surrounded by four connection nodes to build an isotropic material geometry [18]. Heinl et al. investigated the high porosity of diamond cellular structures close to mechanical performance to those of trabecular bone [13]. In addition, the unit cell diamond was designed for biomedical applications by [19,20]. A number of works have studied cellular structure in terms of lattice topology for biomedical applications such as cubic [21] and rhombic dodecahedron [14,22]. Jung et al. found that the octahedron cellular structures had superior flexibility than cubic cellular structures, which is useful for the medical application of nasal implant-shaped (NIS) scaffolds [23]. The aforementioned unit cells have been broadly studied previously, and numerical relationships are introduced for predicting the mechanical performance of cellular structures generated by lattice topology [24]. Babaee at el. predicted the mechanical behavior of rhombic dodecahedron by analytic solution [25].

Additive manufacturing (AM) is the most promising capability of fabricating cellular structures which are in high interest across varying industries such as medical, automotive, and aerospace [26]. AM enables to fabricate structures with high geometrical intricacies such as these achieved by optimization techniques and without any additional tooling cost over a traditional process, which is in stark to subtractive manufacturing methods. Therefore, AM technology, such as selective laser melting (SLM) are broadly used to manufacture cellular structure Ti-6Al-4V alloys for biomedical application. The SLM process is enabled to fabricate 3D periodic cellular structures via the printing of layered structures based on the CAD model [27,28]. In the SLM process the laser beam was scanned over the powder surface in order to melt and fuse metallic powders [29], and performed with high efficiency and reproducibility [30]. In recent times, a cellular structure manufactured via SLM is considered as a potential applicant for scaffolds and bone implants, because of better mechanical performance [31,32]. Simoneau et al. also produced a porous stem altogether with its fully dense replica parts of the Ti–6Al–4V using SLM [33]. Limmahakhun at el. fabricated a graded cellular for application of scaffold bones using SLM [34]. Furthermore, Hazlehurst et al. manufactured cellular structures that have similar strength and stiffness properties as cancellous and cortical bone of human femur using SLM [28]. The SLM process can fabricate cellular structures based on the type of lattice topology with various relatively new lattice topologies. Therefore, it is more convenient to study the type of lattice topology and unit size because these characteristics are used to determine the mechanical performance of cellular structures [35]. Therefore, it is a key issue to study the effect while varying lattice topology and unit cell size of lattice structures on the mechanical performance. Heinl et al. studied that the permeability and mechanical properties of cellular implants depend on their morphological characteristics such as type of lattice topology, pore size, and porosity [36]. In addition, Van bael et al. studied the biological performance (bone growth, differentiation, cell attachment) of cellular structures that depend on the porosity, pore architecture, and pore size [37]. It is accordingly essential to study the biological performance, mechanical performance, and fluid flow of a various type of lattice topology in order to

develop a library of different unit cell lattice topology which could be utilized for optimal design of scaffolds and implants.

Therefore, many researchers have designed and investigated different types of cellular structure, however, no structure has optimum mechanical properties along with a desirable pore size and porosity that could improve biological performance (bone growth) and mechanical properties which can maintain the comfort of the patient. Consequently, new structures are needed to be designed, manufactured, and compared with already existing lattice structures in order to achieve better performance particularly for biomedical applications. In our previous paper [38], the vintiles cellular structure was compared to the mechanical performance with the existing cellular structures such as cubic, hexagonal, rhombic dodecahedron. According to the FEA and experimental testing results the mechanical performance of the vintiles type cellular structure were found as having better mechanical performance than the above mentioned cellular structures. In addition to that, presently there are no studies with vintiles lattice topology design in the implant and scaffold and no numerical relationships for analyzing the mechanical performance of cellular implant made of the vintiles lattice topology.

The objective of this study was to design, simulate, optimize, fabricate, and mechanically test AC cellular implant incorporating a new vintiles lattice topology with pore size ranging 600–1200 μm to investigate the overall stiffness reduction and to calculate the yield strength, ultimate compressive strength, and young's modulus. AC cellular implant and their conformity with the similar features of the host bony tissue. These features reveal the potential to mimic the matching structural characteristics of host bony structures and decrease the stress-shielding phenomenon.

## 2. Materials and Methods

### 2.1. Material

Cellular titanium alloys are promising biomaterials for achieving bone-mimicking. AM techniques, for example, SLM are able to fabricate cellular titanium alloy structures with the accurate design of micro-architecture [39]. In addition, titanium alloys are highly degradable, corrosion-resistant, high-strength, low-density material and this material has a wide-ranging use in biomedical industry [40–42]. Mainly, the Ti6Al4V powder was used in several studies due to its excellent biocompatibility. Therefore, due to the above reasons, titanium alloy powder was chosen in this study. The Ti-6Al-4V powder was bought from the electro-optical system (EOS) GmbH, Germany. Figure 1 shows the SEM micrograph original titanium alloy powder at different magnifications and the chemical composition of Ti-6Al-4V is listed in Table 1.

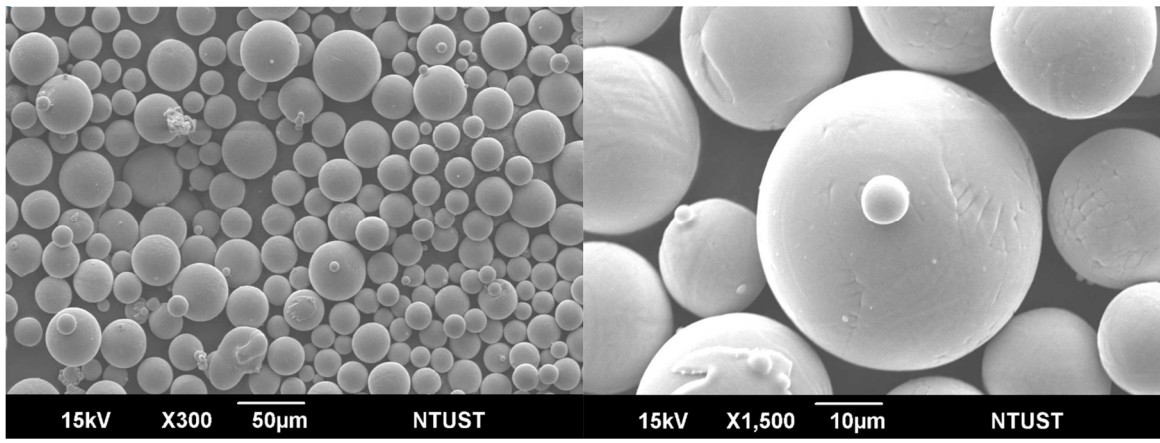

**Figure 1.** SEM micrograph of Ti6Al4V powder at different magnification.

**Table 1.** Chemical composition of Ti6Al4V alloy powder used for SLM.

| Element | Ti | Al | V | O | N | C | H | Fe |
|---------|------|------|------|------|------|------|---------|------|
| % | (balance) | % | % | % | % | % | % | % |
| Wt | Balance | 6 | 4 | <0.2 | <0.5 | <0.08 | <0.0125 | <3 |

## 2.2. Unit Cell Selection

The biological and mechanical properties of a unit cell size for a periodic cellular biomaterial are mainly dependent on types of lattice topology, porosity, pore size, and the monolithic material from which they are built [43,44]. In the previous study, six different types of lattice topology with different unit cell sizes were used to compare their mechanical performance, therefore, vintiles cellular structure was selected as the optimum design [38] see in Figure 2a–d vintiles lattice topology with different thickness 0.4 mm, 0.8 mm and 1 mm. Moreover, vintiles lattice topology fulfills Maxwell's criterion for static determinacy, indicating that it is stretch dominated for all loading states. Based upon the lattice topology of the unit cell, cellular materials are generally classified into two main groups, namely, stretch dominated and bend dominated [45]. The unit cells are a category to the former collapse by the stretching of their struts, a failure mechanism that provides a higher stiffness, and strength per unit mass as compared to bend dominated topologies. A unit cell belonging in the latter group collapses by the local bending of the cell struts at the nodes thereby, leading to lower mechanical properties. For orthopedic applications, a cellular biomaterial should have the adequate mechanical strength to withstand a combination of physiological loadings. Hence, this is the reason stretch dominated cell topologies are selected to design a cellular biomaterial with higher porosity for bone ingrowth and sufficient mechanical properties.

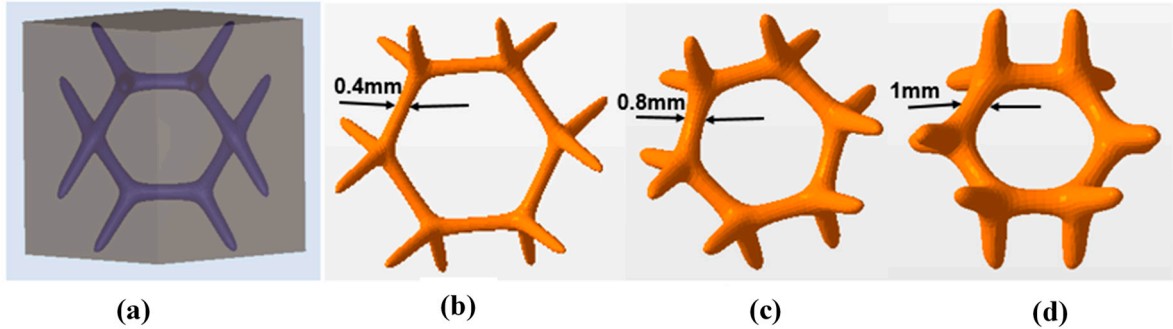

**Figure 2.** Unit cell with different struts. (**a**) Unit size in block (**b**) with 0.4 mm beam thickness, (**c**) 0.8 mm beam thickness, (**d**) 1 mm beam thickness.

## 2.3. Design of Ti-6Al-4V Alloy Implant (AC) with Vintiles Structure

Following the CT(computed tomography) scan (Figure 3a) before segmentation of the acetabular cup (AC) from the pelvis to obtain the two-dimensional (2D) CT images of AC, the 2D data and Mimics 16.0 software (Materialise Inc., Leuven, Belgium) is used to rebuild the three-dimensional (3D) model of the AC (see the procedure in [46]). After we constructed the 3D AC in Figure 4a, two different unit cells size were taken for generating AC cellular implant used with a vintiles lattice topology. These unit cells are 5 × 5 × 5 and 6 × 6 × 6 mm with different porosity and pore size as seen in Figure 5 and Table 2 in all four samples. To introduce the vintiles unit cell lattice topology into the solid CAD (computer aided design) model, which is the AC implant in this work, the vintiles lattice topology and the AC solid implant was used to transform the solid CAD model into an AC cellular implant. Thus, the unit cell of the cellular implant structure should be designed first, the vintiles unit cell was automatically formed since the unit cell size and strut thickness were inputted into the Autodesk Netfabb software in this work, as shown in Figure 4. Then, a different cellular implant with an expected porosity controlled

by unit cell and strut of thickness was designed. All the cellular implants were designed as AC with a size 52 (52 × 36 mm), as shown in Figure 4c.

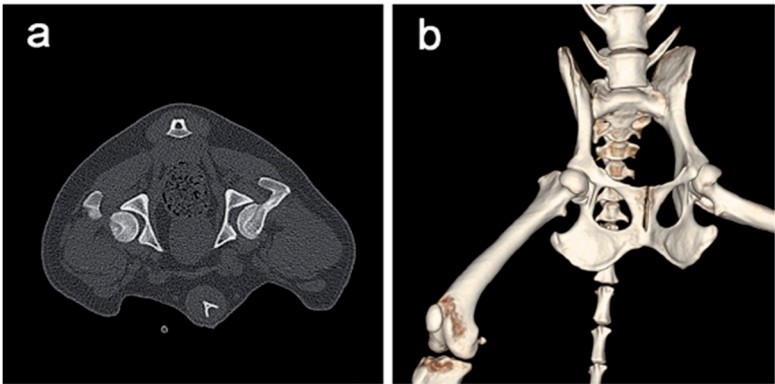

**Figure 3.** CT scan image. (**a**) CT scan images before segmentation; (**b**) three-dimensional (3D) reconstruct from CT scan [47].

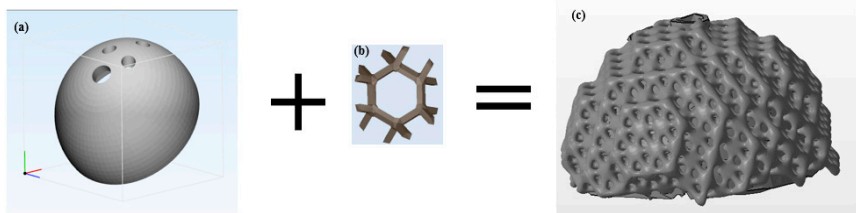

**Figure 4.** The schematic illustration showing the design process of cellular implant structure. (**a**) Solid AC CAD, (**b**) vintiles lattice topology, and (**c**) cellular implant AC.

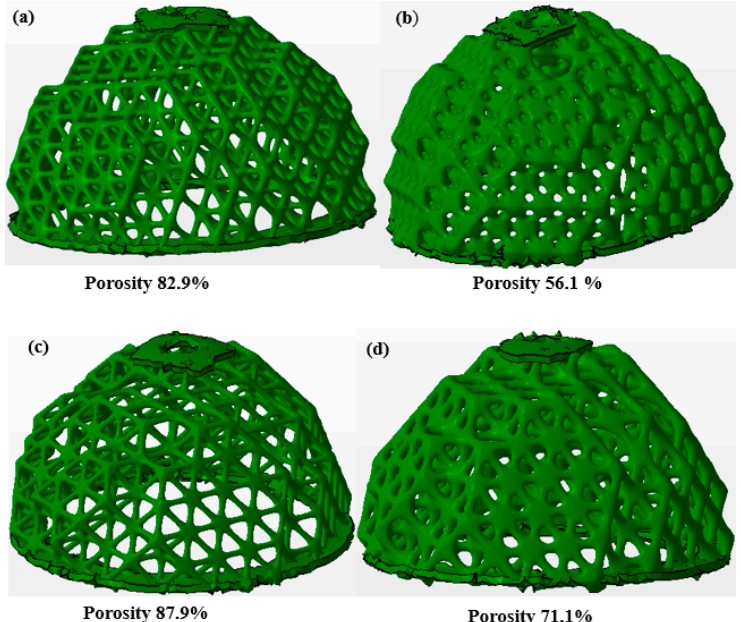

**Figure 5.** 3D CAD model of AC with varying unit cell size and porosity. (**a**) 5 × 5 × 5 mm non-optimized, (**b**) 5 × 5 × 5 mm optimized, (**c**) 6 × 6 × 6 mm non-optimized, and (**d**) 6 × 6 × 6 mm optimized.

### 2.4. Porosity of AC Implant

The porosities of all AC cellular implants are analyzed using the relationship of both volume of cellular implants, as well as the original volume (solid implant). The scientific formula for determining the porosity ρ of cellular implants was calculated using the following formula Equation (1).

$$\rho = 1 - \frac{\text{Volume}_{cs}}{\text{Volume}_{solid}} \tag{1}$$

where

ρ is the porosity of the implant, $\text{Volume}_{cs}$ is the volume after introducing the cellular structure, and $\text{Volume}_{solid}$ is the volume before cellular structure was introduced.

The range of porosity during analysis is from "0" to "1". If the analysis is close to "1" that means that the implant is more porous, while if the analysis is close to "0" value that means the implant is dense.

**Table 2.** Cellular structures for compression testing.

| Cellular Structure | Unit Size (mm × mm × mm) | Strut Size Max. (mm) | Pore Size (mm) | Porosity (%) |
|---|---|---|---|---|
| Non-optimized | 5 × 5 × 5 | 1.3 | 1154 | 82.9 |
| Optimized | 5 × 5 × 5 | 0.6–1.3 | 654 | 56.1 |
| Non-optimized | 6× 6 × 6 | 1.3 | 1214 | 87.9 |
| Optimized | 6 × 6 × 6 | 0.6–1.3 | 974 | 71.1 |

### 2.5. Finite Element Analysis (FEA)

FE simulation was performed to study the effect of porosity and pore size on compression mechanical performance of the cellular implant of AC for ideal cases where the material is assumed to be isotropic. For that purpose, the CAD model were designed and imported into the Netfabb software (Figure 6a). The material property of an implant model is shown in Table 3. These material properties are consistent with the base powder used to manufacture the experimental implant samples. According to Affatato et al., applied forces and boundary conditions were used to generate an implant model that simulates the loading force on a physical structure [48]. The triangular mesh was generated, and the mesh was refined until stresses converged, with a final average mesh size of approximately 100,000 elements (see Figure 6a,b). Through the use of FEA on a combined model, an acetabular cup implant can be simulated in a software system. This predicts an alternative to physical testing using an actual implant by stress and deformation which is more resource and time-consuming.

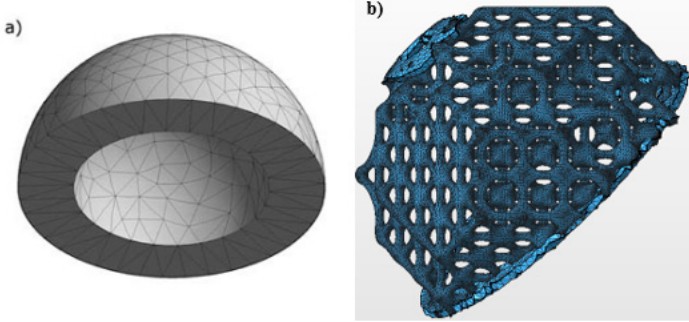

**Figure 6.** Meshing on an AC implant model for FEA, (**a**) solid and (**b**) cellular implant.

**Table 3.** The material properties of Ti-6Al-4V used in the FEA [49].

| Material | Young's Modulus | Poisson's Ratio | Compressive Yield Strength | Tensile Yield Strength |
|---|---|---|---|---|
| Ti-6Al-4V | 108 ± 1 GPa | 0.36 | 1187 ± 5 MPa | 1320 ± 6 MPa |

## 2.6. Fabrication of AC via SLM

Cellular implants with varying unit cell sizes, porosity, and optimization (variable density) were fabricated by the SLM machine (SLM-125, NAR Lab Taipei, Taiwan) equipped with a 200 WCW ytterbium fiber laser. All fabrication processes took place in a protective argon atmosphere in which $O_2$ is less than 0.1 vol%. The process parameters used in this work are listed in Table 4. Twelve AC cellular implants with different porosity and unit cell sizes fabricated by the SLM process are shown in Figure 7. The fabricated cellular implant was removed from the base plate by cut off using wire electrical discharge machining (wire-EDM) for various tests.

**Table 4.** Process parameter of SLM used to obtain the cellular implant.

| Processing Parameters | Value |
|---|---|
| Laser power | 95 W |
| Scanning speed | 1250 mm/s |
| Distance between laser scanning lines | 40 μm |
| Laser Spot diameter | 100 μm |
| Scan spacing | 75 mm |
| Volumetric energy density | 45.3 J/mm$^3$ |

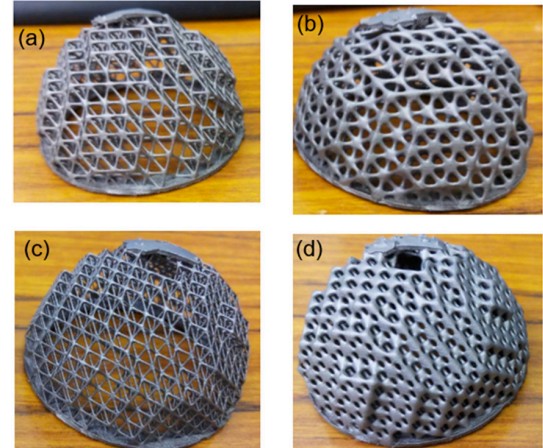

**Figure 7.** Some of the cellular implants (AC) with vintiles structures built by the SLM process. (**a**) Non-optimized with 5 × 5 × 5 mm unit size, (**b**) optimized with 6 × 6 × 6 mm unit size, (**c**) non-optimized with 5 × 5 × 5 mm unit size, and (**d**) optimized with 5 × 5 × 5 mm unit size.

## 2.7. Mechanical Property Cellular Implant AC

To verify the simulation results, the experimental testing was performed based on the ASTM standard D1621-10 to determine the mechanical performance of AC cellular implants. The compressive testing was conducted using a universal compression testing machine. The compressive load for the fabricated AC cellular implants were applied along the build orientation, as shown in Figure 7. The strain rate of the compression load was set to 0.1 mm/min and the load is applied up to the total strain of 20 mm in the plastic deformation range. The compressive stress and strain relationship were obtained using the following formulas Equation (2).

$$\{\sigma\} = E\{\varepsilon\} \tag{2}$$

where $\{\sigma\}$ is the stress vector, $\{\varepsilon\}$ is the elastic strain vector, and E is the Young's Modulus of Ti6Al4V alloy. It can be assumed that theTi6Al4V material is an isotropic material. Therefore,

$$E = E_x = E_y = E_z \text{ and } \nu = \nu_{xy} = \nu_{yz} = \nu_{zx} \tag{3}$$

where $\nu$ is the Poisson's ratio of the Ti6Al4V material.

*2.8. Optimization Process of the Cellular Implant of AC*

Even though cellular structures are advantageous in weight, naturally they are non-optimal in regard to structure configuration and material usage. Many studies [50–54] have been investigated to optimize the unit cell components and shape within a cellular structure in order to achieve the design requirements of the given object. The final generated object is governed by an objective function constrained by constraint equations and variable value limitations. Several methods are developed for optimizing cellular structures with each technique giving priorities to external inputs in order to develop the most suitable design cellular structure to the given function. To enhance the performances of periodic cellular structures in Autodesk Netfabb software, there are two approaches; the first one is the optimization utility package, which can be used to optimize the cellular structure based on skins and lattices of various sizes, densities, and thicknesses to create superior 3D parts and the second one is the topology optimization package that can optimize the shape of cellular structure or solid 3D component itself by the computational method. In this study, we used the first approach optimization unity package to optimize the lattice structures. The optimizing process was carried out by FEA software tools, which are used to simulate and optimize the capability of the designed cellular structure components output. It was also supported by combining with the Autodesk Nastran FEA solver. We carried out optimization tasks using the following process. The objective optimization in this study was to design a light-weight implant by minimizing the volume while the stress constraint must be met as per final requirements of allowable stress. The optimization algorithm removes the weight from less stressed regions whereas it adds material in the regions of high stresses. In this way, it makes the part stiffer in some areas and flexible in others.

i.      Importing the AC solid implant, then introducing vintiles lattices topology into AC solid implant with the desired unit size and thickness of strut.

ii.     Then, define loads and boundary conditions for the component designed to optimize and by defined variable density of lattice thickness with min–max value (0.4–1.3 mm).

iii.    Defined lattices and skins based on load and boundary conditions then validate the performance of components by FEA.

iv.     According to the defined load and boundary conditions, manually changing the density of lattices topology and skins in components to increase until we achieve the goal of our design.

v.      Then, automatically optimize the lattice topology of defined cellular parts using the optimizing Autodesk Nastran to come across the defined boundary conditions and constraints. See the process and flow chart of the optimization of the lattice component in the previous work [38].

## 3. Results and Discussion

*3.1. Manufacturability of Vintiles Cellular Structure on the Implant by SLM*

The acetabular cup implant with vintiles lattice structures with different porosity (optimized and non-optimized) and unit cell sizes 5 and 6 mm were manufactured using the SLM process with no obvious deformations. The models of the AC implant with vintiles lattice topology structures and having different unit cell sizes are shown in Figure 7.

When the unit cell size is increased as the beam thickness is being constant, it results in the unit cell numbers to decrease in AC samples with a thinner strut connecting them. Though the unit cell size reduced to a small size, the struts become very small so it is very challenging to remove the unnecessary powder after the fabrication of cellular implant via SLM process. In this work, since the unit cell size was well suited to the manufacturing constraint of SLM, the unnecessary powder was entirely removed (see the fabricated samples in Figure 7). The smallest unit cell size of the cellular implant which can be fabricated by the SLM process generally depends on laser beam diameter of the SLM and the metal powder particles sizes. If the smaller laser beam diameter and finer metallic powder are set as the parameter process on SLM, a smaller unit cell can be fabricated with thinner struts.

In contrast to the above, as the unit cell size decreased with the beam thickness being constant, the total number of unit cells in the cellular implant increases, as well as increases in the volume reduction coefficient as the beams become shorter and stronger. In contrast, the deformation increases as the beam length becomes longer during the fabrication process of cellular implant. The cellular implant sample with the unit cell size of 6 × 6 × 6 mm was fabricated without distortion, a lower volume reduction coefficient, and has a lower number of unit cells compared to other cellular implants. In contrast to the cell size, 5 × 5 × 5 mm was fabricated with a higher number unit cell and increase in volume reduction. During the SLM process, residual stresses developed due to high thermal rises in the interior material which are generated by rapid melting, high T (°C), and cooling melting. Osakada et al. demonstrate that the failure causes cracking and distortion to happen due to the residual stresses [55]. On the other side, researches have been done on the effect of angle of beams built on the fabrication process of SLM leading to deformation of the cellular structure. Among these, santorinaios et al. investigated that for the cellular structure with beams (struts) of a certain degree angle from the horizontal plane, deformation will occur and more support material is required for their fabrication, furthermore, the cellular implant that has a beam with a certain degree angle and with a unit cell size greater than 7 mm is difficult to be fabricated because of the resulting deformation [56].

Figure 8 shows that SEM images of the beams of the implant are built by SLM. The beams in the cellular implant structure built by SLM are circular in the cross-sectional area and the cores are spherical as per designed, that is similar to the CAD design shown in Figure 5a. Figure 8b shows a higher magnification of SEM morphology cross-sectional area of beams, even if the morphology shows that there is no balling effect on the surface, the powder is totally melted but the observed cracks with an approximate length of 15 μm and thin width less than 0.5 μm. Many researchers that studied the happening of these fractures (cracks), revealed that it is caused due to high internal stress distribution (residual stress) through the SLM manufacturing process [55]. As shown in Figure 8c, the bonded particles which are not fully melted are found on the surface of the beams cellular implants. Both Pattanayak et al. and Santorinaios et al. investigated the same phenomena in the beams of cellular implant samples with different metallic powder Ti6Al4V and stainless steel [56,57], respectively. However, in this study, the cause of this phenomenon has not been discussed. The presence of bonded (unmelted) particles on the surface of the beams may be caused due to the partial melting happening and the balling occurrence of raw metallic powder particles on the surface of the beams. An advanced feature of the SLM process such as direct metal laser sintering (DMLS) has a high potential to the entirely melted powder without no effect of balling [58]. However, in Figure 8b SEM images show that there is no observing balling phenomenon on the cross-sectional of the beams surface. Therefore, the observed bonded particles cannot appear because of the resulting balling phenomenon. The bonded particles observed in Figure 8c in higher magnification of the SEM images have a spherical geometry and their size is approximately 45 mm and the coarse surface follows fine particles and a sporadic shape adhere on it. When compared with the morphology between the observed bonded particles in the surface of beams (Figure 8c) and metallic powder TiAl4V (Figure 1b) have similarity in size. These additional phenomena indicate that these bonded particles were not produced due to the balling effect but due to the partial melted metallic powder on its boundary part.

Since the bonded particles have an effect on the beams surface by increasing their surface roughness cellular structures. Therefore, to remove such roughness from the surface of the structure, a small size particle powder had better use, with the post-processing such as sandblasting can be painstakingly done to clean the bonded particles from the surface of the beams structure this is aiding the damage of the thin beams during the blasting process. Furthermore, Pattanayak et al. investigated that the fully melted and bonded partially melted Ti6Al4V particles with core parts have been used by heat treatment process in an argon atmosphere at 1300 °C [57].

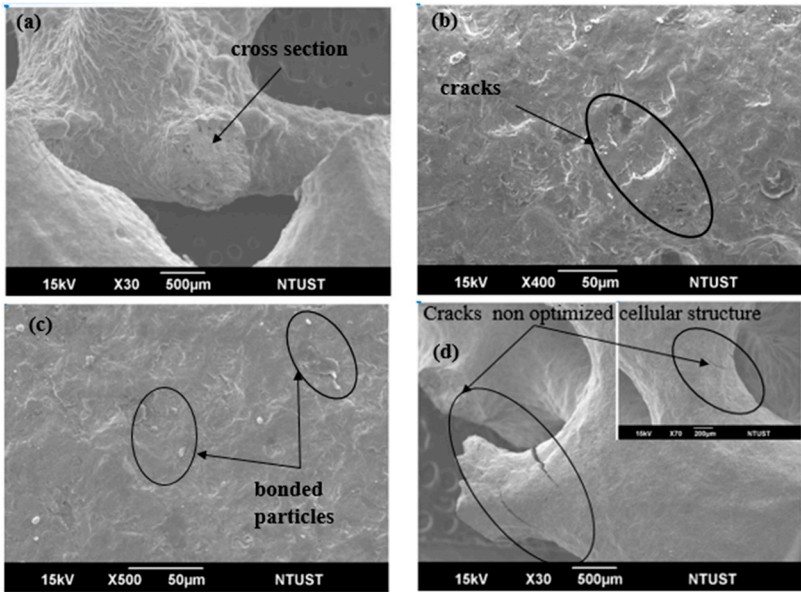

**Figure 8.** SEM images of (**a**) the cross-section cellular implant with vintiles cell size of 4 × 4 × 4 mm (**b**) the cracks on the surface of beam, (**c**) bonded particles on the surfaces of the beams, and (**d**) cracks on non-optimized surface.

## 3.2. Simulation Result

The most essential parameters for the cellular structure to be used in the orthopedic implants is the effect of stiffness and strength of cellular implant. In static load simulations using homogenous materials, elastic parameters were used to compare stress distributions and Young's modulus in varying unit cell size, non-optimized, and optimized cellular implant designs. Therefore, to verify the validity of the mechanical performance the FEA method was carried out in this work. The FEA for each designed non-optimized and optimized cellular implant is successfully completed and a maximum equivalent stress and the displacement results are given in Table 5. As expected, maximum VM stress and displacements results are directly proportional to the unit cell size, the porosity, and pore size on the cellular implant structures. The increasing pore size and porosity mainly lead to larger deformation. It is significant that the displacements of the cellular implants are in the elastic region. Maximum VM stresses and high displacement occur in the upper region of all cellular implants. These results show that it is similar to the ones introduced by Sivasankar et al. and Delikanli et al. [59,60]. As shown in Table 5 and Figure 9, the less porosity cellular implant (optimized) contains the smallest maximumVM stresses (533.81 MPa) compared to the higher porosity cellular implant. In contrast, higher porosity cellular implants experience much higher stresses under the same boundary and loading conditions since the cross-sectional pore size is higher. In addition, the maximumVM stresses optimized and non-optimized cellular implant simulation results data are analyzed and the values of non-optimized and optimized unit cell size 5 × 5 × 5 mm are 1.71 GPa and 533.81 MPa, respectively which means a comparative difference of 69%. On the other hand, the maximum VM stresses of the non-optimized and optimized with unit cell size 6 × 6 × 6 mm is evaluated and the value 5.16 and 1.92 GPa for the simulation, which leads to a comparative difference of 63%. In the simulation results, the overall stress decrease provided by the increasing unit cell size and porosity, as compared to its small unit cell size and less porous counterpart, is greater than 40%.

**Table 5.** FEA results.

| Cellular Implant (AC) Unit Cell Size (mm) | Porosity % | Pore Size (mm) | Strut Diameter (mm) | Max. Von Mises Stress (MPa) | Displacement (mm) |
|---|---|---|---|---|---|
| 5 × 5 × 5 non-optimized | 82.9 | 1154 | 0.7–1.3 | 1710 | 0.29 |
| 5 × 5 × 5 optimized | 56.1 | 654 | 0.7–1.3 | 533.81 | 0.09 |
| 6 × 6 × 6 non-optimized | 87.9 | 1214 | 0.7–1.3 | 5160 | 0.96 |
| 6 × 6 × 6 optimized | 71.1 | 974 | 0.7–1.3 | 1920 | 0.3 |
| solid | 0 | - | - | 72.47 | 0.009 |

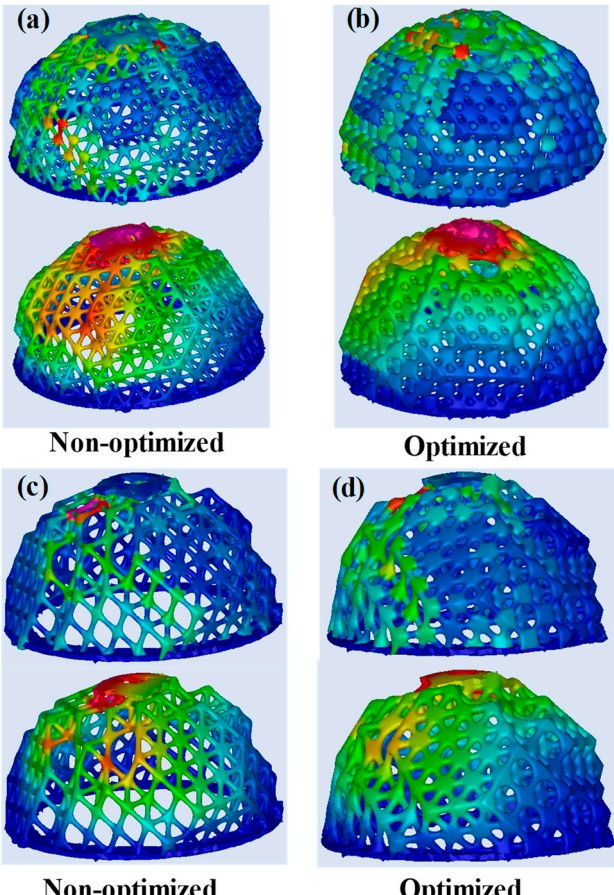

**Figure 9.** Results of FEA (**a**) unit cell size 5 × 5 × 5 mm non-optimized, (**b**) 5 × 5 × 5 mm optimized, (**c**) 6 × 6 × 6 mm non-optimized, and (**d**) 6 × 6 × 6 mm optimized.

### 3.3. Mechanical Properties of the Implant

The effects of unit cell size on the mechanical performance of the AC cellular implant is presented in Figures 10, 11 and Table 6. The influence of the pore size, porosity, and unit size on the stiffness, compressive yield strength, and compressive ultimate strength of the cellular titanium implant are shown in Figure 10a,b, respectively. The yield strength, Young's modulus, and ultimate compressive strength are decreasing linearly with the increasing unit size and porosity. As the results show, the maximum value of yield strength 110 MPa is achieved for the AC cellular titanium implant with a porosity of 56.1% and the yield strength value for others of AC cellular titanium implant with the porosity of 71.1%, 82.9%, and 87.9% are reduced linearly to 85, 65, and 55 MPa, respectively. In addition, the cellular implant with 56.1% porosity, the values of its Young's modulus is 18 GPa and for the cellular implant with the porosity of 71.1%, 82.9%, and 87.9% the value of Young's modulus decrease to 13, 8, and 5 GPa, respectively. Young's modulus values are comparatively low as compared to a solid implant (120–130 GPa) [61]. The manufactured cellular titanium implant characterizes a relatively

higher UCM (ultimate compressive strength). The UCM of the cellular titanium implant with the total porosity of 60%, 75%, 85%, and 90% are 140, 100, 65, and 35 MPa, respectively. According to the results, the value of the compressive ultimate strength and Young's modulus decrease with the increasing of pore size and porosity, and the value of modulus elasticity is between 5 and 13 GPa, which is in the range between 0.3–20 GPa of modulus elasticity human bones.

Moreover, all the AC fabricated sample masses were measured by using a precision electronic (digital) mass balance to calculate the exact lightening rates. In Table 7 the measurement results are shown, the lower mass which is lighter AC cellular implants was found with a lightening value rate of 78.07%, 36.44%, 81.25%, and 57.79% for unit cell size $5 \times 5 \times 5$ mm non-optimized, $5 \times 5 \times 5$ mm optimized, $6 \times 6 \times 6$ mm non-optimized, and $6 \times 6 \times 6$ mm optimized, respectively. Additional reducing in mass of cellular implant might bring stability and strength issues, while the increase in mass could have a negative effect on the bone tissue growth at the implantation area. Therefore, it is essential to consider these circumstances when designing cellular implants. In future, the authors would investigate the fatigue, and biomechanical properties of AC implants, as well as hip implant. The hip implant needs to be lighter in weight without compromising the biomechanical strength.

**Table 6.** Results of experimental work.

| Unit Size mm × mm × mm | Porosity % | | Pore Size (mm) | | Peak Load (N) | | Displacement (mm) | |
|---|---|---|---|---|---|---|---|---|
| | Non-Opt. | Opt. | Non-Opt. | Opt. | Non-Opt. | Opt. | Non-Opt. | Opt. |
| $5 \times 5 \times 5$ | 82.9 | 56.1 | 2.031 | 0.964 | 7894 | 24,517 | 2.5 | 3.1 |
| $6 \times 6 \times 6$ | 87.9 | 71.1 | 2.568 | 1.538 | 3879 | 13,239 | 1.9 | 3 |
| solid | 0 | | 0 | | 11,5225 | | 3.3 | |

**Table 7.** Sample implants mass and reducing rates.

| Samples | Mass (g) | Reducing Weight Rate % |
|---|---|---|
| $5 \times 5 \times 5$ mm non-optimized | 12.88 | 78.07% |
| $5 \times 5 \times 5$ mm optimized | 37.33 | 36.44% |
| $6 \times 6 \times 6$ mm non-optimized | 11.012 | 81.25% |
| $6 \times 6 \times 6$ mm optimized | 24.79 | 57.79% |
| Solid | 58.73 | - |

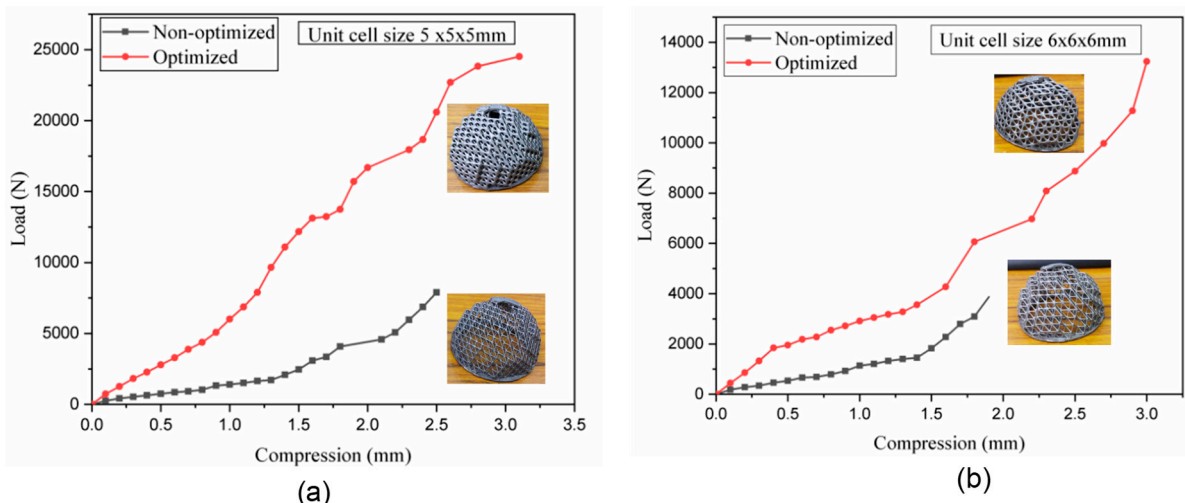

**Figure 10.** Compressive load-compression curves for optimized and non-optimized (**a**) unit cell size with $5 \times 5 \times 5$ mm, (**b**) unit cell size with $6 \times 6 \times 6$ mm.

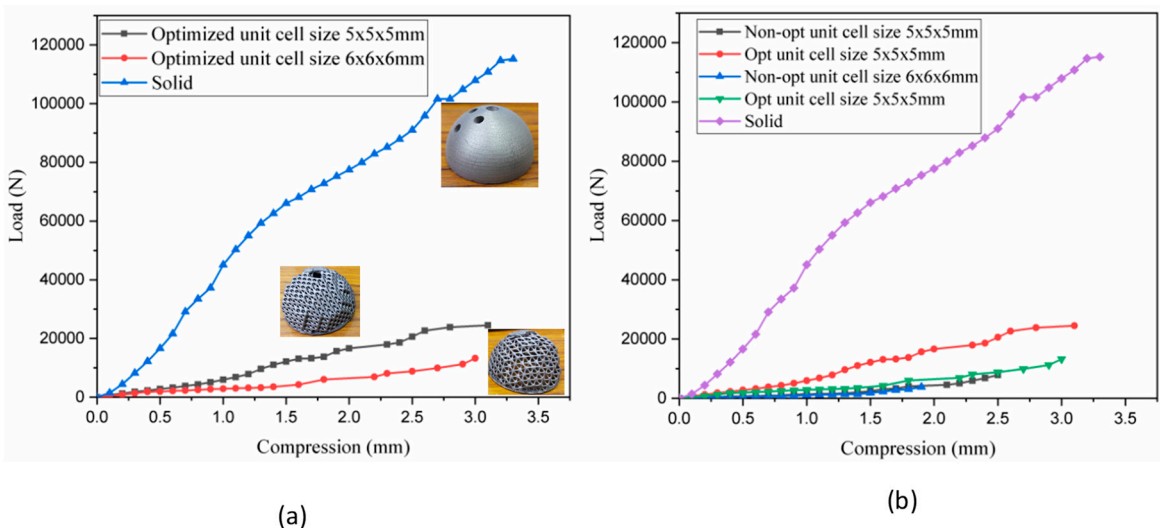

**Figure 11.** Compressive load-compression curves. (**a**) Solid with optimized, (**b**) all solid, optimized, and optimized.

### 3.4. Optimization Results

As shown in Figure 10a,b the comparison of optimized and non-optimized cellular implant with different unit cell sizes, as the graph shows the optimized in both unit cell size, has a much higher stiffness than the non-optimized of both unit cell sizes, indicating that the porosity of the non-optimized AC cellular implant quickly increased, and the peak stress of the non-optimized AC cellular implant significantly reduced, that means the volume reduction of the non-optimized cellular implant reduced sharply, and the strength of the cellular implant was significantly decreased. Figure 11a,b shows the comparison among solid and optimized and non-optimized cellular implants, as shown in the graph the solid implant has a much higher stiffness than both cellular implants indicating that both cellular implants (optimized and non-optimized) linearly decrease in stiffness as they increase the porosity. Therefore, in this study, the optimization process was more effective. In order to achieve better stress values, lighter were achieved in the optimized cellular implant. Though, decreasing stress values also result in decreasing peak stresses. Both contradicting goals of designing an AC cellular implant, which is both strong and lightweight, make it difficult to decide which structure is better between the two optimized cellular implants. Eventually, for best security, the cellular optimized implants with stress value 533.81–5160 MPa was selected for final comparison of the structure, because they attain a better balance between porosity and stability. As a final point, compared with the solid cellular implant, the optimized AC cellular implant achieved a 36.44% reduction in weight.

### 4. Conclusions

In this paper, authors have presented the design and optimization approach for, and investigated the performance of vintiles structure for a biomedical AC implant, which is a component of the total hip prosthesis. It is revealed that a lighter weight with low stiffness of the AC implant as compared to its solid counterpart, can be designed by introducing a regular vintiles lattice structure, which also improved the bone ingrowth. The results show that the proposed AC cellular implant with a vintiles lattice topology cellular implant has high porosity, good interconnectivity, and is stable enough to resist normal loads. Using the proposed optimization method, the optimized cellular implant has reduced the stress shielding and has a more uniform stress distribution. The maximum VM stresses of all the non-optimized and optimized AC cellular implant is evaluated and the optimized AC cellular implant is reduced averagely by 66% compared to non-optimized. The weight of the optimized AC cellular implant is reduced by 69% than the solid one. The modulus elasticity of both optimized AC cellular implant value is between 5 and 13 GPa, which is comparable to 0.3–20 GPa of the modulus elasticity

of human bones. In addition, the porosity and pore size of the optimal cellular implant decreased, which ultimately improved the mechanical performance of the implant, and also improved the stability similar to the actual bone. Furthermore, by optimizing the pore size, unit cell size, and the porous structure with consideration of practical requirements, a perfect match of mechanical properties can be achieved for AC implants, as well as other biomedical implants.

The study found that the SLM is capable of direct fabrication of AC implants with expected mechanical properties. By using the AM technologies, cellular implants can be fabricated with tailored mechanical properties, using comparatively less amounts of material as preferred by medical doctors.

**Author Contributions:** Conception and design of study: J.-Y.J., K.M.A.; Supervision: J.-Y.J.; analysis and/or interpretation of data: K.M.A.; Methodology: K.M.A., Drafting the manuscript: K.M.A. revising the manuscript critically for important intellectual content: A.N.; Experimentation: K.M.A., contribute in resources: J.-E.C., Approval of the version of the manuscript to be published: K.M.A., A.N., J.-E.C, J.-Y.J. All authors have read and agreed to the published version of the manuscript.

**Funding:** This research was funded by Jeng-Ywan Jeng grant number 108P012. And The APC was funded by High speed 3d printing Research center.

**Acknowledgments:** This work was financially supported by the High-Speed 3D Printing Research Center from the Featured Areas Research Center Program within the framework of the Higher Education Sprout Project by the Minister of Education (MOE) Taiwan.

**Conflicts of Interest:** The authors declare that there is no conflict of interest.

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
