# Peer review of "Design, Optimization, and Evaluation of Additively Manufactured Vintiles Cellular Structure for Acetabular Cup Implant"

_processes, doi:10.3390/pr8010025_

Round 1

Reviewer 1 Report

The present work shows that the SLM is capable for the direct production  of AC implants with expected requirements.

The work is very interesting and very well written. However, I have very few comments. 

In Figure 2 the labels are completly unreadable. In Table 4 you report the process parameters. How do you select this combination of process parameters?  Line 409. you speak about "normal loads" referring to the comparison between the solid AC cellular implant and the others. What do you mean with "normal loads"? Can you express it with a value? The solid AC cellular implants is the one currently used in surgery? if it is so, I suggest to report the main mechanical characteristics, at least the values that you want to test and compare to strongly support your results.  There some minor errors and typos.

Author Response

Dear Editor, and Reviewers

Thank you for giving us the opportunity to submit a revised draft of my manuscript titled “Design, Optimization and Evaluation of Additively Manufactured Vintiles Cellular Structure for Acetabular Cup Implant” to the Processes. We appreciate the time and effort that you and the reviewers have dedicated to providing your valuable feedback on my manuscript. We are grateful to the reviewers for their insightful comments on my paper. We have been able to incorporate changes to reflect most of the suggestions provided by the reviewers. We have highlighted the changes within the manuscript.

Here is a point-by-point response to the reviewers’ comments and concerns. The reviewer comments are in italics.

Comments from Reviewer 1

The present work shows that the SLM is capable for the direct production of AC implants with expected requirements.

The work is very interesting and very well written. However, I have very few comments. 

Comment 1:   In Figure 2 the labels are completely unreadable.

Response:  Thank you for pointing this out.  As suggested by the reviewer, the existing figures 2 have been replaced with high resolution figures 2.

Comment 2:   In Table 4 you report the process parameters. How do you select this combination of process parameters? 

Response:  We collaborate with a research center in Taiwan to fabricated the samples for this study. The research center has more than 20 years of 3d printing experience for various industrial sectors. Based on their wide experience, we selected these process parameters. The research center has been cited in the study. 

Comment 3:   Line 409. you speak about "normal loads" referring to the comparison between the solid AC cellular implant and the others. What do you mean with "normal loads"? Can you express it with a value? The solid AC cellular implants are the one currently used in surgery? if it is so, I suggest to report the main mechanical characteristics, at least the values that you want to test and compare to strongly support your results. 

Response:  Thank you for pointing this out. As we described in line 409. The results show that the proposed AC cellular implant with vintiles lattice topology cellular implant has high porosity, good interconnectivity, and stable enough to resist normal loads. Here normal load means the load which applied on AC during static condition. And also we are explained the value mechanical characteristics such as yield strength, stiffness and ultimate compressive strength of AC compare to human bone.

Comment 3:   There some minor errors and typos. 

Response: Thank you for concern. As suggested by the reviewer, we tried our best to correct each minor typos errors. We have highlighted the references in the revised manuscript.         

Reviewer 2 Report

Dear Authors,

The aim of this study was to design new acetabular cup implant with porus structure and investigate those mechanical properties. While the topic is fitting to the journal scope, some major concerns were raised. Revise the manuscript by following comments.

Major points

There was no purpose in the Abstract section. Add the clear purpose of this study by following the purpose described in the Introduction section.

What was the objective variable for optimization? Maximum principal stress/strain, volume of implant, healing period? Add the clear explanation.

The end of the Introduction section contained conclusion and future works. Remove them or move them to appropriate section.

Ti6Al4V was assumed as orthogonal anisotropic material. However, an isotropic material properties were defined by following equation 3. Add the clear reason.

The optimization process is not clear. The authors might use a function in software. However, this process is important to understand why stress concentration could be reduced. Add the explanation.

The Results and discussion section should be separated into the Results section and the Discussion section for better readability.

The Conclusion section is too long. Summarize them. Future works must be moved to the Discussion section.

 Minor points

Abstract

"Osseointegration" should be modified to "osseointegration".

"focuses" should be modified to "focused".

"vin tiles" should be modified to "vintiles".

"non-optimized and" is the typo? Revise the sentence.

Introduction

"The objective of this study is" should be modified to "The objective of this study was".

"optimization" should be modified to "optimizing".

Materials and Methods

"duet to" should be modified to "due to".

Table 3

Some standard deviations were lacking. Standard deviation is not required as the material properties for FEA.

Section 2.8

"components by" is typo? Revise the sentence.

Results and discussion

"unite cell" should be modified to "unit cell".

"1.92Gpa" must be modified to "1.92GPa".

Table 6

"gram" should be modified to "g".

Author Response

Dear Editor, Reviewer

Thank you for giving us the opportunity to submit a revised draft of my manuscript titled “Design, Optimization and Evaluation of Additively Manufactured Vintiles Cellular Structure for Acetabular Cup Implant” to the Processes. We appreciate the time and effort that you and the reviewers have dedicated to providing your valuable feedback on my manuscript. We are grateful to the reviewers for their insightful comments on my paper. We have been able to incorporate changes to reflect most of the suggestions provided by the reviewers. We have highlighted the changes within the manuscript.

Comments from Reviewer 2

The aim of this study was to design new acetabular cup implant with porous structure and investigate those mechanical properties. While the topic is fitting to the journal scope, some major concerns were raised. Revise the manuscript by following comments.

Major points

Comment 1:   There was no purpose in the Abstract section. Add the clear purpose of this study by following the purpose described in the Introduction section.

Response:  Thank you for the reviewer concerns. The objective of this study is to design and fabricate Acetabular Cup (AC) implant using a recently designed cellular topology named vintiles structures. Furthermore, the implant was optimized by in order to obtain a lightweight but strong enough implant that can bear the actual loading applied on the AC implant. The objectives have been highlighted in the revised manuscript.

Comment 2:   What was the objective variable for optimization? Maximum principal stress/strain, volume of implant, healing period? Add the clear explanation.

Response:  As suggested by the reviewer, the objective of the optimization has been added and highlighted in the revised manuscript subsection 2.8. The objective of optimization here is to design a light weight implant by minimizing the volume while the stress constraint must be met as per final requirement of allowable stress. Optimization algorithm removes the weight from less stressed regions whereas it adds material in the regions of high stresses. In this way, it makes the part stiffer in some areas and flexible in others.

Comment 3:   The end of the Introduction section contained conclusion and future works. Remove them or move them to appropriate section.

Response: Thanks for pointing this out. We have removed and updated the revised manuscript accordingly.

Comment 4:   Ti6Al4V was assumed as orthogonal anisotropic material. However, an isotropic material properties were defined by following equation 3. Add the clear reason.

Response: Thank you very much for pointing such a big typo and we really apologies for this mistake. By mistake, the space between “an” and “isotropic” was removed. It has been corrected in the revised manuscript and has been highlighted. Ti6Al4V is assumed isotropic material.

Comment 5:   The optimization process is not clear. The authors might use a function in software. However, this process is important to understand why stress concentration could be reduced. Add the explanation.

Response: Thank you for your valuable suggestion. The steps of optimization process have been listed in subsection 2.8 and have highlighted in the revised manuscript. The details of optimization process can be seen from one of our previous study (ref. 39) which is already published in International Journal of Advanced Manufacturing Technology.

Comment 6:   The Results and discussion section should be separated into the Results section and the Discussion section for better readability.

Response: Thank you for your valuable suggestion as per your comment it is good when the discussion and results are independent however in our study it is very challenging and time taking to make results and discussion section separately.

Comment 7:   The Conclusion section is too long. Summarize them. Future works must be moved to the Discussion section.

Response:   As suggested, the future work has been moved to discussion section.

Minor points

Comment 8:   Abstract

"Osseointegration" should be modified to "osseointegration".

 "focuses" should be modified to "focused".

 "vin tiles" should be modified to "vintiles".

 "non-optimized and" is the typo? Revise the sentence.

Response:  The sentences and wording has been updated.

Comment 9:   Introduction

"The objective of this study is" should be modified to "The objective of this study was".

 "optimization" should be modified to "optimizing".

Response:  The sentences and wording has been updated.

Comment 10:    Materials and Methods

"duet to" should be modified to "due to".

Response:  The wording has been updated.

 Table 3 Some standard deviations were lacking. Standard deviation is not required as the material properties for FEA.

Response:  Thank you for your comment. The standard deviations have been updated.

Comment 11:   Section 2.8

"components by" is typo? Revise the sentence.

Response:  The wording and sentence has been updated

Comment 12:   Results and discussion

"unite cell" should be modified to "unit cell".

 "1.92Gpa" must be modified to "1.92GPa".

 Table 6

"gram" should be modified to "g".

Response:  The words has been updated

Round 2

Reviewer 2 Report

Dear Authors,

The manuscript was mostly revised by following reviewer's comments. Some minor concerns were remained. Revise the manuscript by following comments.

Minor points

Equation 3

E=Ex=Ey=Ez is correct? Where is the Ex? Make sure them.

Optimization process

As for the iii, the sentence is not completed. Make sure it.

As for the iv, one sentence is too long. Revise it.

Conclusion

Remove the confirmation of the aim of this study and results. Clear conclusion should be indicated here.

Author Response

Dear Editor, Reviewer

First of all, we would like to say thank you so much for your concern gives 2nd review. We are grateful to the reviewers for their insightful comments on my paper for 2nd revision. We have been able to incorporate the minor comments provided by the reviewers. We have highlighted the changes within the manuscript.

Comments and Suggestions for Authors

Dear Authors,

The manuscript was mostly revised by following reviewer's comments. Some minor concerns were remained. Revise the manuscript by following comments.

 Minor points

Comment 1:  Equation 3 E=Ex=Ey=Ez is correct? Where is the Ex? Make sure them.

Response:  Thanks for pointing this out. We have added Ex, and updated the revised manuscript accordingly.

Comment 2:  Optimization process

As for the iii, the sentence is not completed. Make sure it.

As for the iv, one sentence is too long. Revise it.

 Response:  As suggested, the sentences has been updated in both iii and iv.

Comment 2:   Conclusion

Remove the confirmation of the aim of this study and results. Clear conclusion should be indicated here.

Response: As suggested we have removed the confirmation of the aim of this study, and updated the changes in the revised manuscript